# Advancements in Utilizing Natural Compounds for Modulating Autophagy in Liver Cancer: Molecular Mechanisms and Therapeutic Targets

**DOI:** 10.3390/cells13141186

**Published:** 2024-07-12

**Authors:** Md Ataur Rahman, S M Rakib-Uz-Zaman, Somdeepa Chakraborti, Sujay Kumar Bhajan, Rajat Das Gupta, Maroua Jalouli, Md. Anowar Khasru Parvez, Mushfiq H. Shaikh, Ehsanul Hoque Apu, Abdel Halim Harrath, Seungjoon Moon, Bonglee Kim

**Affiliations:** 1Department of Neurology, University of Michigan, Ann Arbor, MI 48109, USA; ataur1981rahman@hotmail.com; 2Department of Biological Sciences, University of Delaware, Newark, DE 19716, USA; rakibsust07@gmail.com (S.M.R.-U.-Z.); somdeepa7@gmail.com (S.C.); 3Biotechnology Program, Department of Mathematics and Natural Sciences, School of Data and Sciences, BRAC University, Dhaka 1212, Bangladesh; 4Department of Biotechnology & Genetic Engineering, Bangabandhu Sheikh Mujibur Rahman Science & Technology University, Gopalganj 8100, Bangladesh; sujaybge@gmail.com; 5Department of Epidemiology and Biostatistics, Arnold School of Public Health, University of South Carolina, Columbia, SC 29208, USA; rajatdas@email.sc.edu; 6Department of Biology, College of Science, Imam Mohammad Ibn Saud Islamic University (IMSIU), Riyadh 11623, Saudi Arabia; maroua.jalouli@gmail.com; 7Department of Microbiology, Jahangirnagar University, Savar 1342, Bangladesh; khasru73@juniv.edu; 8Department of Otolaryngology-Head & Neck Surgery, Western University, London, ON N6A 4V2, Canada; mushfiq.shaikh@lhsc.on.ca; 9Department of Biomedical Sciences, College of Dental Medicine, Lincoln Memorial University, Knoxville, TN 37923, USA; ehsanul.hoqueapu@lmunet.edu; 10DeBusk College of Osteopathic Medicine, Lincoln Memorial University, Harrogate, TN 37752, USA; 11Division of Hematology and Oncology, Department of Internal Medicine, Michigan Medicine, University of Michigan, Ann Arbor, MI 48109, USA; 12Zoology Department, College of Science, King Saud University, Riyadh 11451, Saudi Arabia; hharrath@ksu.edu.sa; 13Department of Pathology, College of Korean Medicine, Kyung Hee University, 1–5 Hoegidong Dongdaemun-gu, Seoul 02447, Republic of Korea; capre@naver.com; 14Korean Medicine-Based Drug Repositioning Cancer Research Center, College of Korean Medicine, Kyung Hee University, Seoul 02447, Republic of Korea

**Keywords:** autophagy, homeostasis, natural compound, hepatocellular carcinoma, cancer therapy

## Abstract

Autophagy, an intrinsic catabolic mechanism that eliminates misfolded proteins, dysfunctional organelles, and lipid droplets, plays a vital function in energy balance and cytoplasmic quality control, in addition to maintaining cellular homeostasis. Liver cancer such as hepatocellular carcinoma (HCC) is one of the most common causes of cancer deaths globally and shows resistance to several anticancer drugs. Despite the rising incidence and poor prognosis of malignant HCC, the underlying molecular mechanisms driving this aggressive cancer remain unclear. Several natural compounds, such as phytochemicals of dietary and non-dietary origin, affect hepatocarcinogenesis signaling pathways in vitro and in vivo, which may help prevent and treat HCC cells. Current HCC cells treatments include chemotherapy, radiation, and surgery. However, these standard therapies have substantial side effects, and combination therapy enhances side effects for an acceptable therapeutic benefit. Therefore, there is a need to develop treatment strategies for HCC cells that are more efficacious and have fewer adverse effects. Multiple genetic and epigenetic factors are responsible for the HCC cells to become resistant to standard treatment. Autophagy contributes to maintain cellular homeostasis, which activates autophagy for biosynthesis and mitochondrial regulation and recycling. Therefore, modifying autophagic signaling would present a promising opportunity to identify novel therapies to treat HCC cells resistant to current standard treatments. This comprehensive review illustrates how natural compounds demonstrate their anti-hepatocellular carcinoma function through autophagy.

## 1. Introduction

The recent exploration to utilize natural compounds (NCs) to modulate autophagy in hepatocellular carcinoma (HCC) marks a significant stride in cancer treatment. Autophagy, a sophisticated and intricately controlled cellular mechanism, plays a crucial role in reacting to a range of stress factors, including lack of nutrients, invasion by pathogens, and stress in the endoplasmic reticulum (ER), thus ensuring the balance and proper functioning of cells [1]. Recent research indicates numerous NCs can affect autophagy via different signaling pathways, underscoring their potential in cancer treatment [2]. HCC is a deadly form of liver cancer, where autophagy may serve as both a promoter and an inhibitor of the disease. This is particularly evident in the regulation of autophagy by the FOXO family of transcription factors, especially FOXO3a, which is often downregulated in cancers. Activation of FOXO3a can induce autophagy by enhancing the transcription of autophagic regulators like BNIP3 and LC3, linking it to the complex interplay of pathways such as mTORC2 and Akt in cancer cell survival [3]. Numerous signaling routes and critical molecules are involved in regulating autophagy induced by drugs, among which the phosphoinositide 3-kinase (PI3K)-protein kinase B (Akt)-mTOR pathway is considered a central regulator in the control of autophagy [4]. The objective of this research is to elucidate the intricate involvement of autophagy in liver cancer, with the intention of filling the existing vacuum in knowledge. Through comprehending this interaction, our goal is to facilitate the advancement of innovative therapeutic approaches that can manipulate autophagy to enhance the treatment of liver cancer.

HCC’s progression is intertwined with intricate molecular mechanisms. These include the phosphorylation of Fis1, which promotes mitochondrial fission and HCC cells migration, the role of CD44 in increasing YAP expression for HCC progression, and the involvement of Wnt/β-catenin signaling in the advancement of diseases [5]. Three types of autophagy are found in HCC: micro-autophagy, macro-autophagy, and chaperone-mediated autophagy (CMA). Each plays a distinct role in cellular degradation and recycling [6]. The autophagy-lysosomal pathway in HCC is increasingly targeted by small molecules that can modulate autophagy at different stages and significantly influence cell survival and death. They function by regulating autophagy-related signaling and processes, offering a novel approach to HCC treatment [7]. Moreover, metabolic reprogramming in HCC cells, characterized by a shift to aerobic glycolysis (Warburg effect), plays a vital role in the disease’s progression. Targeting this metabolic shift is crucial, as it contributes to the rapid growth and proliferation of HCC cells. The regulation of glycolysis in HCC involves several factors impacting drug sensitivity and cancer cell metabolism [5].

This study seeks to address this gap by conducting a comprehensive examination of the processes by which autophagy impacts the course of liver cancer and by investigating novel treatment approaches that target autophagy pathways. This study aims to consolidate the results of recent research and resolve the contradictions in existing literature to provide a clear understanding of the involvement of autophagy in liver cancer. Additionally, it aims to suggest novel approaches for treating this disease via natural compounds. Recently, many NCs, such as curcumin, resveratrol, bufalin, etc., have been recognized as potential autophagy modulators [8]. These NCs, being smaller in molecular size, often exhibit less toxicity and a higher efficiency in autophagy regulation than traditional pharmaceuticals [2]. This recent discovery emphasizes autophagy’s intricate nature in HCC and highlights the promise of employing NCs to target signaling pathways for efficient cancer treatment. The complex molecular processes governing autophagy and its regulation in HCC pave the way for new therapeutic strategies to enhance patient outcomes.

## 2. Molecular Mechanism of Autophagy in HCC

The molecular role of autophagy in HCC is complex and multifaceted, playing a dual role that can either suppress or promote tumor growth, contingent upon the specific circumstances. HCC is prevented by autophagy and by eliminating damaged organelles and proteins that protect liver cells from harm and death. Autophagy promotes the advancement of liver cancer and impedes the effectiveness of treatment. Autophagy helps HCC cells survive after tumor formation.

### 2.1. A Dual Role in HCC

Autophagy, a cellular process critical for maintaining homeostasis, exhibits a dual character in HCC. Depending on the particular conditions surrounding the disease, this complex mechanism may either inhibit or encourage tumor development [9]. In HCC, the dysregulation of autophagy is a critical factor in disease development. It can either enhance cancer cell survival or trigger cell death, influenced by various elements [5]. Autophagy in HCC impacts apoptosis, cell proliferation, and glucose metabolism. Its role in epithelial–mesenchymal transition (EMT) furthers tumor metastasis. Autophagy contributes to stem cell formation in HCC, leading to chemotherapy and radiotherapy resistance. Targeting autophagy can, therefore, hinder tumor growth and metastasis, enhancing therapeutic responses. In HCC, autophagy is regulated through several different pathways, such as STAT3, Wnt, and different types of RNAs (miRNAs, lncRNAs, circRNAs). Modulating autophagy through antitumor agents is suggested as an effective treatment for HCC [5].

### 2.2. Signaling Cascades of Autophagy in HCC

The signaling cascades of autophagy in HCC are complex and involve multiple pathways, each playing a critical role in the regulation of autophagy, which thereby influences the progression of cancer (Figure 1). Some of these pathways are briefly described below.

PI3K/AKT/mTOR Pathway: The primary pathway through which autophagy regulates HCC. The activation of AKT leads to the inhibition of the TSC1/TSC2 complex, which activates mTORC1. mTORC1 plays a pivotal role as an inhibitory regulator of autophagy, and its activation leads to the suppression of the autophagy pathway [10]. Furthermore, the PI3K/AKT/mTOR pathway is influenced by factors like amino acids, nutrients, and PTEN, a phosphatase that can negatively regulate phosphatidylinositol 3-kinase (PI3K) signaling [11].

AMPK-mTOR Pathway: AMP-activated protein kinase (AMPK) is another critical regulator of autophagy [12]. It can either phosphorylate through the mammalian target of the rapamycin (mTOR) pathway or directly phosphorylate Uridine/Cytidine kinase (ULK1), a protein essential for autophagy. AMPK is also involved in how cells react to energy shortages, such as starvation, which inhibits mTORC1 activation and triggers autophagy [13].

IGF-EGFR-MAPK Pathway: IGF and EGFR play a significant role in the initiation of autophagy in HCC through the activation of crucial cell survival pathways, such as the PI3K/AKT, JNK/c-Jun, and Ras/Raf pathways [14].

Wnt-β-Catenin Pathway: The Wnt-β-catenin signaling pathway also plays an essential role in controlling autophagy within HCC. Without Wnt signaling, β-catenin undergoes degradation; however, when Wnt is present, β-catenin accumulates and influences autophagy regulation [15].

p53 Pathway: The p53 pathway also mediates autophagy in HCC, especially under stressful situations, such as UV radiation. p53 primarily triggers autophagy through established pathways, including PI3K-AKT-mTOR and AMPK-mTOR [16,17].

NF-κB Pathway: It also triggers autophagy by elevating the expression of Beclin 1 and various other proteins associated with autophagy. On the flip side, it can suppress the autophagic process by upregulating the expression of inhibitory proteins, such as BNIP3 and Bcl-2/Xl [18]. These pathways demonstrate the intricate network of signaling involved in regulating autophagy in HCC. Grasping the intricacies of these pathways is essential for developing a precise treatment for HCC, as autophagy plays a dual role in cancer development. Thus, targeting these pathways could offer new avenues for effectively treating HCC [19].

## 3. Natural Compounds Targeting Autophagy in Hepatocellular Carcinoma

In the context of hepatocellular carcinoma (HCC), recent research has illustrated the potential use of NCs in targeting autophagic pathways, suggesting the development of novel therapeutic strategies. The following details provide an overview of essential compounds that have been the focus of recent studies:

Curcumin is a vibrant yellow compound extracted from the ‘*Curcuma longa*’ plant, commonly known as turmeric, which belongs to the ginger family of plants. It has been extensively used in Ayurvedic medicine for centuries due to its therapeutic potential. Recently, curcumin has emerged as a focal point of interest within the scientific realm due to its potential capabilities in addressing a range of diseases, notably cancer and, more specifically, HCC. Curcumin exhibits a robust anti-inflammatory property [20] by inhibiting key molecules, such as nuclear factor-kappa B (NF-kB), cytokines, and cyclooxygenase-2 (COX-2). Since chronic inflammation is a risk factor for cancer, including HCC, reducing inflammation and curcumin could potentially lower the risk of cancer development. Curcumin has been found to directly affect cancer cell growth and survival by inducing apoptosis (programmed cell death) and inhibiting the proliferation of cancer cells, including hepatocellular carcinoma cells [20]. Curcumin can also influence the PI3K/Akt signaling pathway, which is pivotal for cell survival, growth, and multiplication.

Curcumin’s ability to downregulate PI3K/Akt signaling can inhibit cancer cell growth and survival (Figure 2). The mTOR pathway is also critical in regulating cellular growth and survival. It is also considered as a crucial regulator of autophagy. A study suggests that mTOR, overexpressed or hyperactivated in several cancer types, reduces autophagy and increases cancer cell survival [21]. Curcumin’s action on mTOR can help restore autophagic processes, potentially leading to cancer cell death. Earlier studies demonstrated that curcumin selectively targets and triggers cell death in thyroid cancer cells, leaving normal epithelial cells unharmed through autophagy via MAPK activation and mTOR pathway inhibition [21]. Curcumin can also promote autophagy under stress conditions, such as nutrient deprivation or treatment with anticancer drugs. In the context of HCC, curcumin-induced autophagy can lead to autophagic cell death, an alternative form of cell death to apoptosis. This process can help in eliminating cancer cells from the body. By modulating these crucial pathways and promoting autophagic cell death, curcumin holds promise in inhibiting tumor growth in HCC [20] and could be a potential candidate for HCC therapy. Further research is needed, especially clinical trials, to fully understand its therapeutic potential and optimize its use in cancer treatment.

Resveratrol: It is commonly found in grapes and berries and has recently gained prominence in cancer therapy research, including HCC treatment. It has been observed to modulate autophagy through multiple signaling pathways. Resveratrol influences various signaling pathways and gene regulation associated with apoptotic and autophagic cell death [22]. This underscores its potential as an effective therapeutic agent in cancer treatments, primarily by influencing autophagy [23]. Mechanistically, resveratrol can activate AMPK pathway and consequently inhibit the mTOR pathway, which is pivotal in promoting autophagy (Figure 2). This ability of resveratrol to induce autophagy is a critical factor in its potential to suppress the growth and proliferation of HCC cells. This discovery positions resveratrol as a compound of interest in liver cancer research [24], offering new insights into the development of cancer therapies [25]. A recent study revealed that resveratrol potentially reduces the secretion of exosomes, which influence HCC progression [26]. This effect was linked to the downregulation of Rab27a, a key molecule in exosome secretion. In addition, the study found that exosomes influenced by resveratrol could curb the harmful traits of HCC cells by inhibiting β-catenin’s nuclear movement and stimulating autophagy, likely mediated by lncRNA SNHG29 [26].

Paclitaxel: A well-known chemotherapy drug also significantly influences autophagic processes in HCC [27]. The utilization of paclitaxel in cancer therapy has been shown to induce autophagy, especially in cancer stem cells [27,28]. Studies suggest combining paclitaxel’s anticancer effects with agents that modify autophagy to potentiate paclitaxel, such as chloroquine as an inhibitor or apatinib as an inducer. Furthermore, the development and application of nanoparticle-based paclitaxel formulations or their analogs, designed to specifically influence autophagic activities, present an innovative approach to augment its therapeutic effectiveness against cancer [27]. Another research study delved into the interplay between circular RNAs (circRNAs) and paclitaxel in the context of HCC and mainly focused on miR-877-5p, which is hypothesized to act as a tumor suppressor in various cancers, including HCC [29]. The study also examined how miR-877-5p interacts with paclitaxel and its implications on HCC treatment [29]. It specifically analyzed the regulatory dynamics of the circ-BIRC6/miR-877-5p/YWHAZ axis in HCC cells, providing valuable insights and outlining a potential therapeutic pathway for treating HCC with paclitaxel [29].

Bufalin: It is derived from traditional Chinese medicine. Bufalin has recently emerged as a promising agent for treating HCC. It is especially noted for regulating autophagy and boosting immune responses [30]. This compound has demonstrated its capability to suppress the growth and metastasis of HCC cells by interacting with multiple signaling pathways. Notably, its combination with sorafenib, a widely used antibody for cancer treatment, shows a synergistic effect, amplifying its therapeutic potential. This combination not only enhances the effectiveness of conventional cancer treatments but also introduces novel approaches in immunotherapy, highlighting bufalin’s versatility in cancer treatment strategies [30]. Bufalin has also been identified for its therapeutic potential in HCC treatment and is essential in modulating immune responses. It achieves this by maintaining a balance between stimulatory and inhibitory receptors on natural killer (NK) cells, and by suppressing the shedding of MICA. This process typically facilitates immune escape. Consequently, bufalin indirectly stimulates NK cells and reinforces NKG2D-dependent immune surveillance, thereby countering the immune evasion mechanisms of cancer cells. This action of bufalin opens new avenues for its application in HCC therapy, offering a novel perspective on integrating traditional medicine components into modern cancer treatment protocols [31].

Epigallocatechin Gallate (EGCG): Epigallocatechin Gallate (EGCG) is a member of the catechin family of protein, and is a prominent bioactive compound found in green tea. It has garnered significant attention within the realm of cancer research, especially regarding liver cancer. Its role in modulating autophagy presents a notable avenue for cancer therapy. EGCG has been suggested to have anti-inflammatory, antioxidant, and anticarcinogenic properties [32]. EGCG has also triggered autophagy in HCC cells by inhibiting the mTOR signaling pathway (Figure 3). In addition, EGCG can activate the AMPK pathway, promoting autophagy. EGCG can also interfere with the binding between Beclin-1 and Bcl-2/Bcl-XL, which is crucial for the initiation of autophagy. EGCG can induce oxidative stress, which is known to trigger autophagy. This oxidative stress can damage cellular components, necessitating their removal via autophagy [32]. In liver cancer cells (HepG2), the EGCG was shown to cause mitochondria and cytoskeleton damage, a decrease in mitochondrial potential, and disruption of normal actin structure [33]. Importantly, EGCG elevates the production of LC3-II, an indicator of autophagy, and induces the formation of LC3-containing cytoplasmic vesicles (autophagosomes). This process leads to the autophagic degradation of AFP (alpha-fetoprotein) aggregates in the HepG2 cells. Furthermore, the use of 3-methyladenine, an autophagy inhibitor, has been shown to reduce LC3 levels and promote the accumulation of cytoplasmic AFP, underscoring the role of EGCG in stimulating autophagy [33]. EGCG inhibits the development and advancement of HCC through diverse molecular actions both in vitro and in vivo. Moreover, EGCG notably improves the effectiveness of chemotherapy, radiotherapy, and targeted treatments, offering a new approach to therapy for HCC [34].

Crocin: It is a natural carotenoid chemical compound primarily found in saffron, derived from the Crocus sativus flower. It is responsible for saffron’s distinctive golden-yellow hue. Research on crocin has highlighted its possible health advantages, encompassing antioxidant, anticancer, and anti-inflammatory effects [35]. Traditional medicine utilized it for multiple therapeutic applications. Recent research on crocin particularly has sought to understand its role in autophagy modulation in hepatocellular carcinoma and highlighting its significance in medicinal studies. Crocin has been shown to suppress the proliferation of HCC cells in a dose- and time-dependent manner, inducing cell death through apoptosis. This process involved early autophagy, marked by increased LC3 puncta and LC3-II. Crocin also affects Akt/mTOR signaling, essential for its autophagic and apoptotic actions. Inhibiting autophagy using 3-methyladenine also suppressed the apoptosis in HCC cells triggered by crocin. This research highlights crocin’s potential in HCC therapy by emphasizing its novel role in inducing autophagic apoptosis [36]. Treatment with crocin and sorafenib, both individually and combined, effectively restores normal liver structure and reduces key gene expressions related to cancer development in hepatocellular carcinoma (HCC). This treatment significantly lowered the expression of critical carcinogenesis genes (TNFα, p53, VEGF, NF-κB) induced by HCC (Figure 3). Combining crocin with sorafenib showed significant improvement in histopathological and inflammatory parameters over individual treatments. Furthermore, both drugs demonstrated synergistic antitumor effects on HepG2 cells, suggesting that the combined therapy not only mitigates liver toxicity but also effectively counters HCC progression and enhances liver functionality [37]. As the demand for effective cancer treatments grows alongside advancements in science and technology, the investigation into saffron, especially crocin, as a potential cancer treatment is becoming an increasingly popular field of study. Crocin shows significant promise in combating tumors. Innovations, such as liposome-based nano-particle delivery systems for crocin are expected to evolve as breakthrough anticancer medications. Future research is anticipated to further validate the antitumor pharmacological mechanisms of crocin [38].

Betulinic acid (BA): It is a naturally occurring pentacyclic triterpenoid, extracted from the white birch tree, and demonstrates various medicinal qualities. Its most notable attribute is its anticancer potential, demonstrated in various cancer types, including melanoma and leukemia. This compound is unique for its ability to selectively target cancer cells, thereby reducing the risk of damage to healthy cells. Additionally, betulinic acid is known for its antiretroviral, antimalarial, and anti-inflammatory effects [39]. Research continues to explore its mechanisms, including the use of nanotechnology for enhanced delivery, to optimize its therapeutic benefits for clinical applications. A study on betulinic acid’s effect on human gastric cancer MGC-803 cells demonstrated significant changes in the cells’ autophagic activity [39]. Specifically, when treated with betulinic acid at concentrations between 10 to 30 mg/L, there was a notable increase in the mRNA levels of two autophagic genes, *LC3-II* and *Beclin-1*, while the levels of *LC3-I* were depleted. This pattern of gene and protein expression suggests that betulinic acid has the ability to induce autophagy in gastric cancer cells, a process that could be pivotal in cancer treatment strategies [6]. Betulinic acid has been recognized in scientific circles as an effective natural compound with the potential to treat cancer. It stands out for its specific targeting of cancer cells, displaying cytotoxicity against them while sparing healthy cells, which makes it an asset in cancer treatment strategies. This selective action against cancer cells is a significant advantage, highlighting betulinic acid’s potential as a therapeutic agent [39]. The effectiveness of BA extends to various human cancer types, with numerous studies focusing on its ability to prevent, suppress, and manage these malignancies. Betulinic acid works by influencing multiple cellular signaling pathways, a crucial aspect in understanding its mechanism at the molecular level. These insights are primarily drawn from preclinical research, which has been instrumental in revealing the compound’s potential and laying the foundation for future clinical applications. The results of the above-mentioned studies emphasize betulinic acid’s versatility as a natural therapeutic option in oncology, indicating its promise for future treatment modalities [39].

Emodin: Emodin belongs to the anthraquinone family, a type of organic compound known for its distinctive structural features. It is extracted from several plants, most notably from the roots and rhizomes of *Rheum palmatum*, commonly known as Chinese rhubarb. Emodin has recently become a focal point in scientific studies for its promising therapeutic benefits due to its anti-inflammatory, anticancer, antiviral, and antibacterial properties [40]. A recent study indicated that emodin showed a potent effect on reducing the growth in HepG2 cells, with its impact intensifying with higher doses and prolonged exposure [41]. The study also suggested that the observed inhibition by emodin not only leads to diminished cell growth but also decreases cellular proliferation by triggering a halt in the cell cycle, mainly at the S and G2/M phases, thus enhancing the process of programmed cell death. Additionally, it has been found that emodin disrupts HepG2 cells’ ability to migrate and invade [41]. A key feature of emodin’s impact on HCC is its role in controlling autophagy, specifically through the degradation of crucial proteins like snail and β-catenin. This activity is a key aspect of how emodin inhibits cell metastasis in hepatocellular carcinoma (HCC), highlighting the interplay between autophagy and the process of epithelial–mesenchymal transition (EMT). The induction of autophagy by emodin leads to the breakdown of these key proteins, and interrupting autophagic flux post-treatment with emodin can trigger a reversal of EMT (Figure 3). The PI3K agonist Y-P 740 notably influenced the phosphorylation levels of GSK3β and mTOR, suggesting that the mechanisms by which emodin triggers autophagy and prevents EMT involvement are the Wnt-β-catenin and PI3K/AKT/mTOR signaling pathways [40]. Emodin shows notable effectiveness against several cancer types, including those of the liver, breast, lung, and colon. Its anticancer mechanisms are characterized by restricting cell growth, prompting cell death, and hindering tumor growth and spread. It notably influences critical cellular pathways, such as PI3K/Akt, MAPK, and NF-κB [41].

Quercetin: It is a naturally occurring flavonoid, present in a wide variety of fruits, vegetables, leaves, and grains. It is known for its antioxidant properties, which help combat free radicals in the body, potentially reducing inflammation and the risk of various diseases. Research has explored quercetin for its possible anticancer properties, especially its ability to reduce cancer cells and promote apoptosis. Its role in modulating autophagy, a process vital for maintaining cellular health, makes it a compound of interest in cancer research and therapy [42]. Quercetin has been identified as a potential inhibitor of HCC cell growth. It works by inducing apoptosis and stimulating autophagy, notably by blocking the AKT/mTOR pathway and stimulating the MAPK pathways [43]. Laboratory studies have shown that quercetin decreases the viability of HCC cells in ways that depend on both the concentration and the duration of treatment. Its capacity to trigger autophagy contributes to its efficacy in combating HCC. These observations emphasize quercetin’s promise as a therapeutic candidate for HCC, owing to its dual mechanism of apoptosis induction and autophagy stimulation [42]. Quercetin was identified as one of the key active ingredients in *Artemisia rupestris* L., exerting anticancer activity against HCC via the PI3K/Akt signaling pathway. In vitro results reveal that this substance can hinder the proliferation, induce apoptosis, and reduce the migration and invasion of HepG2 cells [43]. A study on *Prunella vulgaris* L. (PVL) explored quercetin’s efficacy in treating HCC [44]. Utilizing network pharmacology and in vitro methods, the study revealed that these components, particularly quercetin, hinder HCC cell growth and encourage cell death, possibly via the PI3K/AKT pathway [44]. This highlights PVL, especially quercetin, as a promising candidate for HCC therapy

## 4. Recent Advances in the Use of Natural Compounds in Liver Cancer

Natural compounds derived from animals, plants, or other natural origins have been widely used to treat many human diseases [45,46,47]. They are still the primary source for more than 60% of the drugs used to treat various types of cancers and continue to be important ingredients in the development novel therapies to fight against cancers [48,49]. Recent studies have provided substantial evidence elucidating their mechanism of action, identification of biological targets and metabolic regulation, further establishing them as promising candidates for cancer treatment [4,28,50]. Furthermore, herbal extracts and bioactive compounds derived from traditional medicines may be used to treat patients with liver cancer [51,52,53].

Multiple functional studies conducted over the past few decades have confirmed that apoptosis initiated in response to cell-death signals, serves as a natural barrier to the progression of cancer [54]. It has provided evidence demonstrating that natural compounds can activate one or more of these signaling pathways to facilitate cell death [55,56]. A recent report indicates that bioactive compounds like psoralen derived from *Psoralea corylifolia* can trigger apoptosis in liver cancer cells [57]. Natural compounds can also arrest the cell cycle and disrupt cell cycle regulatory processes to stop the growth of cancerous cells in liver cancer [58]. They suppress the migration and progression of HCC by regulating numerous signaling pathways like MKK4/JNK, PI3K/Akt/mTOR, RAF/MEK/ERK, and WNT/β-catenin [59,60,61] (Figure 4).

Over the past fifteen years, increasing evidence indicated a beneficial role of curcumin in inhibiting the invasion and migration of liver cancer cells. CUR3d, an analog of curcumin, at a concentration of 100 μmol/L, can prevent liver cancer cell growth by suppressing the NF-κB and PI3K/Akt pathways [62]. Moreover, a supplement of curcumin also inhibited the growth and metastasis of both liver and colorectal cancer cells when applied at a concentration of 1g/kg [51]. Resveratrol, another highly efficient anticancer compound against HCC, functions by suppressing the PI3K/Akt signaling, thereby triggering autophagy and preventing the progression of HCC [63,64]. Additionally, it decreases the malignancy of HCC by elevating the expression of phosphatase and tensin homolog (PTEN) [63]. Resveratrol also reduces the population of CD8^+^ CD122^+^ Treg cells, thus promoting antitumor immunity in HCC [65]. Moreover, it enhances the expression of p53, leading to cancer cell growth inhibition in HCC [66]. Another recent study confirms that by downregulating membrane-associated RING-CH (MARCH 1), resveratrol suppressed PTEN/Akt pathway and induced apoptotic cell death in HepG2 and Hep3B cells [67]. Notas et al. demonstrated that even a short treatment with resveratrol interrupted DNA replication and arrested the cell cycle [68]. In an in vivo study, resveratrol when administered intraperitoneally at a concentration of 1 mg/kg was also showed to arrest the cell cycle and suppressed tumor growth in AH-130 hepatoma cells implanted in Wistar rats [51]. In a recent study, Rajasekaran et al. revealed that resveratrol exhibited chemopreventive properties in a DEN-induced HCC model in Wistar rats (male) [69]. This was achieved through the induction of apoptosis, characterized by PARP cleavage, activation of caspase-3, upregulation of p53, and release of cytochrome c [69]. MHCC97H-inoculated nude athymic mice also revealed the antitumor activity of resveratrol by downregulating the HGF/c-Met signaling pathway [70]. Moreover, to improve its efficiency, resveratrol-gold nanoparticles have been developed with improved anticancer effects and have demonstrated such in HEPG2 cells and Balb/c nude mice, respectively [71].

Early research suggests that crocin, a chemical constituent of saffron, may potentially exert beneficial effects on HCC. Research has demonstrated the potential of crocin to effectively hinder the spread and advancement of HCC [37]. Crocin works by inducing autophagic apoptosis in HCC through the inhibition of AKT/mTOR activity [35]. It can also prevent STAT3 activation in Hep3B and HepG2 cells induced by IL-6 [72]. The inactivation of STAT3 in liver cancer cell lines was induced by the inactivation of *JAK1*, *JAK2*, and Src kinase [73]. Moreover, STAT3 can be dephosphorylated by expressing SHP-1, a protein tyrosine phosphatase (PTP) [73]. Another study demonstrated that crocin has shown an anti-proliferative effect in HepG2 cells by arresting the progression of S and G2/M phases in the cell cycle through the initiation of apoptosis and the downregulation of the inflammatory responses [74]. In addition, crocin acts on the interleukin (IL)-6/STAT3 pathway through the inhibition of JAK1/2 and Src kinases in Hep3B and HepG2 cells [75]. Yao et al. also found that crocin in an AKT/mTOR-dependent way can activate autophagy, which leads to the induction of crocin-mediated apoptosis [35]. Moreover, to improve the efficacy and drug availability of crocin, Awad et al. combined it with saffron, a multi-kinase inhibitor, which showed higher anticancer effects in the cirrhotic rat model of HCC by downregulating the NF-κB-p65, COX-2, and β-catenin [37].

Like crocin, emodin also has shown to suppress cell proliferation, arrest the cell cycle, and trigger apoptosis via PI3K-AKT pathway in HCC [37]. It has also been shown to prevent EMT by modulating the interaction between the Wnt-β-catenin pathway and autophagy in HepG2 cells [40]. Furthermore, emodin can regulate macrophage polarization towards either M1 or M2 and the invasion of HepG2 cells by enhancing the secretion of TGFβ1 [76]. Zhu et al. demonstrated that a derivative of emodin, aloe-emodin, decreased the expression of PI3KR1, AKT1, and BCL2 mRNA levels in HCC through the downregulation of the PI3K-AKT pathway [77].

Natural compounds like betulinic acid (BA) and quercetin can also inhibit the Akt/mTOR pathway to trigger autophagy and induce apoptosis. Chen et al. showed that BA inhibited the growth of HCC, PLC/PRF/5, and MHCC97L cells by inducing both apoptosis and autophagy. [78]. Liu further validated that BA inhibited cellular growth and triggered apoptosis and autophagy by impeding the PI3K/AKT/mTOR signaling [79]. Wang also demonstrated that BA impeded proliferation in HCC cell lines, such as HepG2, LM3, and MHCC97H [80]. Moreover, BA can inhibit pulmonary metastasis in HCC by altering metastatic proteins such as MMP-2, MMP-9, and TIMP2 [43]. Similarly, Tu et al. illustrated that quercetin could inhibit cellular growth and trigger apoptosis in HCC cells by depleting the expression of downstream genes associated with the PI3K/AKT pathway, such as *PI3K, AKT, GLUT4* and *IRS-2* [81]. Another study by Luo et al. unveiled that quercetin can suppress the invasion, migration, and proliferation of HCC cells through increased apoptosis, which is facilitated by the upregulation of PTEN expression and activation of PTEN/AKT pathway [82]. In addition, Licochalcone A, a *Glycyrrhiza inflata* root-derived compound, was suggested to inhibit the invasion and migration of HCC cells by decreasing the expression of MKK4/JNK signaling [83].

There have been advances in understanding how natural compounds act on newly discovered therapeutic targets for liver cancer treatment [84]. Davidone C is a recently identified flavonoid molecule that is extracted from *Sophora davidii*, which is referred to in the study that investigated its antitumor efficacy against HCC cells [85]. The anti-HCC effects of platycodin D are achieved via inducing autophagy. They can explain that it triggers the activation of the ERK and JNK signaling pathways, which result in the creation of autophagic vacuoles and higher levels of LC3-II protein [86]. Parthenolide is a naturally occurring chemical present in some plants that possesses properties of reducing inflammation and fighting against cancer. Research has demonstrated that parthenolide has the ability to trigger autophagy in HCC cells, resulting in the demise of the cells [87]. Polyphyllin D, an inherent constituent present in botanical species such as the *Podophyllum peltatum* L., has demonstrated notable anticancer capabilities. Research indicates that it can affect autophagy in HCC cells in various ways [88,89]. Sarmentosin stimulates the autophagic process in HCC cells. The mTOR pathway is a significant suppressor of autophagy. Sarmentosin enhances autophagy via blocking mTOR [90]. Shikonin is a bioactive chemical present in certain plant species that exhibits promising anticancer characteristics. Research indicates that it may have an impact on the treatment of HCC via affecting the process of autophagy [91]. Solamargine suppressed the growth of liver cancer cells and triggered apoptosis and autophagy in both laboratory trials and animal models by influencing a unique signaling pathway that involves LIF, miR-192-5p, CYR61, and Akt [92].

The significance of natural chemicals in regulating autophagy and their possible therapeutic benefits in liver cancer has garnered significant attention. This discussion includes detailed information about each chemical, with a specific emphasis on doses, administration routes, mechanisms of action, and potential side effects, which are all presented in Table 1. Turmeric-derived curcumin has been extensively explored for anticancer potential. Oral dosages of 100–200 mg/kg suppress mTOR signaling, which regulates autophagy. Common side effects include stomach distress [93]. Grape resveratrol stimulates the AMPK system and promotes autophagy at 10–100 mg/kg orally. Although safe, users may experience headaches and dizziness [94]. Berberine from goldenseal causes autophagic cell death at 5–50 mg/kg orally. Careful dosage monitoring is needed due to nausea and constipation [95]. Onions contain quercetin, which inhibits the autophagy-regulating PI3K/Akt pathway at 25–50 mg/kg orally. Users may get headaches and stomachaches [96]. EGCG, found in green tea, promotes autophagy via modulating Beclin-1 and Bcl-2 at 50–200 mg/kg orally. However, large amounts may damage the liver [97]. Oral genistein from soybeans suppresses Akt/mTOR signaling at 10–50 mg/kg. The most prevalent side effects are gastrointestinal [98]. Lycopene from tomatoes induces autophagy via AMPK at 10–50 mg/kg orally. There are few allergic responses and little toxicity [99]. Apigenin in parsley suppresses PI3K/Akt at 10–40 mg/kg orally. Possible adverse effects include mild stomach pain [100]. Baicalein, from *Scutellaria*, controls autophagy and apoptosis at 10–50 mg/kg orally. Use may cause nausea and vomiting [101]. Honokiol from magnolia activates AMPK and suppresses mTOR at 5–50 mg/kg orally or intravenously. It may cause sleepiness [102]. Silibinin from milk thistle suppresses mTOR and activates AMPK at 100–300 mg/kg orally. Common side effects include mild stomach disturbances [103]. Oral withaferin A from ashwagandha regulates the p62/SQSTM1 pathway at 5–20 mg/kg. Side effects include nausea and redness [104]. Rhubarb contains emodin, which inhibits the PI3K/Akt/mTOR signaling pathway at 10–40 mg/kg orally. Diarrhea and stomach ache can occur [105]. Diosgenin, a fenugreek-derived compound, suppresses Akt/mTOR signaling at 10–50 mg/kg orally. Common side effects include mild stomach pain [106]. Oral plumbagin from black walnut promotes autophagy via ROS at 2–10 mg/kg. Effects may include hemolysis and nephrotoxicity [107]. Ursolic acid in apple peel suppresses mTOR signaling at 10–50 mg/kg orally. It usually causes mild gastrointestinal issues [108]. Oral fisetin from strawberries suppresses the PI3K/Akt/mTOR pathway at 10–50 mg/kg. Mild digestive difficulties are common [109]. Oral luteolin from celery suppresses Akt/mTOR signaling at 10–50 mg/kg. Possible adverse effects include mild stomach pain [110]. Ginsenoside Rg3, produced from ginseng, inhibits mTOR and promotes autophagy at 5–30 mg/kg orally or intravenously. Some users report slight stomach discomfort [111]. Oral capsaicin from chili peppers activates AMPK at 2–10 mg/kg. Common side effects include gastrointestinal discomfort [112].

Natural chemicals can regulate tumor cell development by fine-tuning apoptosis and autophagy communication [113]. Analysis examines natural compounds influencing autophagy as a treatment for liver disease. It critically evaluates current research to identify key approaches and proposes improvements for future studies [114]. Understanding the molecular pathways of autophagy regulation and liver cancer development may spur translational investigations that lead to new liver cancer treatments [115]. Nevertheless, the involvement of autophagy in cancer is intricate and has been linked to both the promotion and the suppression of tumor growth. Several artificial autophagy regulators have been recognized as potential possibilities for the treatment of cancer [116]. Therefore, the molecular relationships and mechanisms for new therapeutic methods in the targeting and modulation of autophagy in hepatocellular cancer must be examined [117].

## 5. Preclinical and Clinical Evidence of NCs Directing Liver Cancer via Autophagy

Evaluating the pharmacological effectiveness of phytochemicals depends on their capacity to suppress tumor growth in animal models. An increasing amount of evidence suggests that NCs are capable of inducing autophagy and effectively can suppress tumor growth *in vivo* [118,119]. However, few clinical trials have been conducted in liver cancer utilizing autophagy-inducing natural compounds up to this point. Several natural products have been found to be effective in treating liver cancer, such as resveratrol, which showed potent chemopreventive effects [120]. Clinical trials have confirmed that resveratrol has a beneficial effect on cancer patients, whether administered alone or in combination with chemotherapy drugs [121]. Several studies have indicated that the consumption of resveratrol during a phase I clinical trial involving healthy volunteers did not lead to significant adverse side effects [122].

Resveratrol’s ability to combat liver carcinogenesis and its potential in anticancer applications were demonstrated through reduced incidence and diminished nodule numbers in various animal models (transgenic mice, including those expressing the hepatitis B virus X protein (HBx)) exposed to chemical carcinogens such as DENA (diethylnitrosamine), DENA and phenobarbital, and DENA and 2-AAF (2-acetylaminofluorene) [70,123]. In addition, resveratrol was found to elevate the activity of quinone reductase (QR) enzyme in Hepa 1c1c7 cells (mouse liver cancer cell line) [124]. Other studies have also proposed that resveratrol might hold promise in HCC cancer chemotherapy, particularly in combination with other drugs, primarily because of its impact on apoptosis [125]. Another study showed that sorafenib (Sor) displayed greater inhibition in tumor growth when combined with resveratrol (Res) compared to treatment with Res or Sor alone in vivo [126].

Numerous research yielded promising results regarding the efficacy of curcumin in treating liver cancer, particularly HCC. Curcumin has shown promise in treating HCC due to its various pharmacological effects targeting the disease [127]. According to data from multiple clinical trial databases, curcumin boosts the effectiveness of chemotherapy and radiotherapy, resulting in enhanced expression of antimetastatic proteins and a decline in associated side effects [128]. In addition, a water-soluble derivative of curcumin, 3,5-bis(2-hydroxy benzylidene) tetrahydro-4H-pyran-4-one glutathione conjugate [EF25-(GSH)2], induced autophagy at a concentration of 5 μM and at 10 μM, in a human hepatic cell line, HL-7702 [129]. Chen et al. (2022) illustrated that curcumin administered at doses spanning from 10 to 40 μg/mL in HCC and HCCLM3 and Huh7 cell lines, and 200 mg/kg/day in an animal model, showed to effectively target Vimentin, circ_0078710, MMP-9, and Bax, leading to induce apoptosis and inhibit cell growth and migration [130]. In another study by Bai et al. (2022), curcumin at concentrations of 10, 20, 40, and 60 μM targeted various cellular components including Cyt-C, GSK-3β, p-GSK-3β, Caspases 3/9, p-PI3K, Bcl-2, Bax, Akt, p-Akt, and BCLAF1, and led to cell cycle arrest, induced apoptosis, and inhibited cellular growth observed in immunodeficient mice, and in HepG2 cell lines [131]. Additionally, the β-cyclocitral-derived mono-carbonyl curcumin analog A19, administered at concentrations of 5, 10, and 20 μM, affected various cellular components including caspase-3, CDC2, Cyclin B1, MDM2, cleaved-PARP, JNK, ERK, Bcl-2, Bax, and p38 [132]. This intervention may lead to apoptosis induction, cell cycle arrest, and suppression of colony formation in HepG2 and Huh-7 cell lines [132].

Yin and Xiong (2022) showed that crocin significantly reduced blood urea and serum creatinine levels, along with malondialdehyde, while elevating glutathione peroxidase, glutathione, superoxide dismutase, and catalase levels during lipid peroxidation in cisplatin-treated cancer cells [133]. Another study by Abdu et al. (2022) revealed that crocin administered either individually or in conjunction with sorafenib, notably improved the microscopic liver pathology by mitigating degenerative changes and suppressing the proliferation of HepG2 cancer cell lines [134]. Furthermore, Bao et al. (2023) also showed that crocin, when administered at a concentration of 20 µM, can affect Bax and IL-6/STAT3, resulting in apoptosis induction and inhibition of cellular growth in both HepG2 and Hep3B cells [38]. According to the experimental findings reported by Xiu et al. (2023), BA inhibited HCCLM3 and HUH7 liver cancer cell growth by promoting ferritinophagy [135]. In animal trials, a dosage of 20 mg/kg of betulinic acid was administered every 2 days, with the human equivalent dose approximately tenfold higher than that used in mice [135]. In addition, a new delivery system consisting of silver nanocolloids loaded with BA exhibited increased cytotoxicity in both A549 and HepG2 cell lines compared to the original compound BA_DMSO, resulting in significantly reduced cell viability and altered morphology [136].

A preclinical study has demonstrated that emodin can induce apoptosis in hepatocellular carcinoma cells and increase cancer cell death by triggering the expression of TNF-α [137]. In a separate study by Yin et al. (2022), it was observed that treating M2 macrophages with varying concentrations (0, 25, and 50 μM) of emodin in a co-culture system with Huh7 and HepG2 cancer cell lines resulted in a notable decrease in both the proliferation and invasion compared to the control group (0 μM) by promoting M1 macrophages polarization [138]. Moreover, in a preclinical investigation, it was demonstrated that quercetin arrests the S and G1 phases of the cell cycle, and triggers apoptosis in HepG2 cells [134]. This effect was observed within a concentration range of 20 to 220 µM, leading to a reduction in cancer cell proliferation [134]. In a separate investigation, quercetin and Permethylated Anigopreissin A (PAA), acting as inhibitors of hGDH1, triggered apoptosis in HepG2 cells [139]. This effect occurred at concentrations of 1, 2, 4, 8, 12, and 16 μM, leading to a decrease in Bcl-2 anti-apoptotic proteins and mitochondrial mass, along with an increase in mitochondrial superoxide anion and Caspases 3/7/9 [139]. In addition, a preclinical study indicated that Polyphyllin I extracted from *Paris polyphyllin* rhizoma also exhibited anticancer properties via autophagy in certain cancers, including HCC [140].

## 6. Current Challenges and Future Directions for the Use of Autophagy-Mediated Natural Compound in Liver Disease

Currently, the application of drugs targeting autophagy is restricted due to their lack of specificity and potential off-target effects, with no available means to regulate autophagy [28,141]. This article outlined the existing information on NCs that inhibit liver cancer by modulating the autophagy pathway, along with an overview of the ongoing clinical trials involving these compounds. The potential drawbacks associated with these NCs could pose challenges to their advancement in clinical drug development. Potential drawbacks associated with NCs include the presence of inactive metabolic by-products, limited aqueous solubility, inadequate intrinsic activity, suboptimal absorption, and excessive metabolism. Natural compounds also exhibit diverse bioactivities, indicating their interaction with numerous molecular proteins [9,142]. This lack of specificity may limit their effectiveness in clinical applications and can even lead to toxicity.

A Phase II clinical trial with curcumin demonstrated that elevated doses of curcumin can induce gastrointestinal discomfort, skin inflammation, and chest tightness in advanced pancreatic cancer patients [143]. Additional evidence indicated that the use of curcumin at 8g/day dosage might be linked to certain adverse effects [144]. The study also showed similar results when extended administration of curcumin for 1 to 4 months, at doses from 0.9 to 3.6 g/day, led to adverse side effects, such as vomiting, diarrhea, nausea, increased serum levels of alkaline phosphatase and lactate dehydrogenase [144]. A recent study suggested that resveratrol has direct interactions with over 20 proteins. The study further explains the numerous side effects and diverse bioactivities associated with resveratrol, contributing to its lack of specificity [145]. Betulinic acid (BA), although having various health advantages such as selective antitumor properties, also encounters limitations in its in vivo utilization because of its insufficient water solubility and bioavailability [136]. In addition, challenges such as reduced oral bioavailability and inadequate aqueous solubility undermine quercetin’s suitability as a dependable option for therapeutic applications in various cancer types, particularly in liver cancer. Finally, despite the positive pharmacological activities associated with emodin, its exploration is impeded by the observation of hepatotoxicity, nephrotoxicity, and reproductive toxicity. Initial studies revealed emodin’s hepatotoxic effects [146], with further research by Chen et al. indicating cytotoxic effects in hepatocytes and the suppression of hepatocyte nuclear factor 4α expression. This led to reduced expression of UDP-glucuronosyltransferase 2B7 (UGT2B7) and subsequent hepatotoxicity, particularly in cases of long-term or high-dose emodin exposure [147]. Wu L. et al. also highlighted gender differences in emodin-induced hepatotoxicity and toxicity, potentially mediated by the in vivo coupling of UGT2B7 and multidrug-resistant-protein 2 [148].

The use of NCs to modulate autophagy for treating liver cancer can be facilitated by several approaches, for example, by using nanoparticles (NPs), and miRNAs or by altering the structure of these compounds for better specificity and bioavailability. Nanoparticles prevent medications from degradation and improve targeting to treat HCC more effectively. Nanodrug delivery technologies can boost the immune response, making them promising for HCC immunotherapy [149]. The utilization of nanostructures for the modulation of autophagy and treatment of HCC could enhance their prospective future application, which has already been proven in a number of studies. Research on nanoparticles (NPs) targeting autophagy for liver cancer treatment is currently underway, which has also highlighted the regulatory role of silver NPs in autophagy to prevent cancer spread [150,151]. These silver NPs have the potential to deliver chemotherapeutic agents and autophagy regulatory proteins, and ultimately offering a synergistic approach to cancer therapy. Furthermore, gold and zinc oxide NPs have demonstrated the ability to enhance apoptosis and autophagy mediated by oxidative stress, thereby reducing cancer progression [152]. For instance, gold-loaded polymeric NPs were shown to block autophagosome-lysosome fusion, induce autophagy, and promote cell death in breast cancers by inhibiting thioredoxin reductase and increasing ROS levels [153]. In colorectal cancer cells, iron NPs accumulate in mitochondria, disrupting glucose metabolism and inducing autophagy [154]. A number of recent studies have proposed the utilization of nano-sized delivery platforms as strategic approaches to improve the bioavailability and effectiveness of BA [155]. Several formulations, including carboxylic acid-functionalized carbon nanotubes, poly(lactic-co-glycolic acid)-loaded nanoparticles, bovine serum albumin (BSA)-poly(L-lactic acid) nanoparticles, gelatin-based double-targeted nanoparticles and chitosan-coated iron oxide nanoparticles have been suggested to enhance the antitumor efficacy of BA [155,156]. Moreover, quercetin-conjugated metabolites, specifically aglycone-conjugated quercetin metabolites, are synthesized with the help of enzymes, allowing effective delivery to various tissues. Nanocarriers for quercetin and its active metabolites have been identified to enhance solubility in water, prolong circulation time, improve absorption rates, and increase target specificity [145]. Future directions in autophagy modulation for cancer treatment involve overcoming the limitations of natural compounds through advanced delivery systems like nanoparticles (NPs).

Moreover, genetic tools such as miRNAs, small interfering RNAs (siRNAs), short-hairpin RNAs (shRNAs) and CRISPR/Cas9 system can be employed to control autophagy. For example, miRNA-4535, identified within melanoma stem cells, suppresses autophagy and fosters metastasis [157]. Another microRNA, miRNA-99b-5p, regulates through the mTOR/AR signaling pathway to trigger autophagy, and inhibit the proliferation of pancreatic cancer cells [151]. Scientists have altered curcumin’s chemical structure by eliminating the β-diketone group, resulting in a notable increase in the activation of autophagy [158]. Furthermore, chemical modifications of specific autophagy modulators, considering their chemical structures, also present an opportunity for more precise and efficient drug development.

## 7. Conclusions

The autophagy system in normal and malignant cells has diverse functions depending on the situation. Therefore, to maintain homeostasis in a physiological context, the activation of autophagy at a baseline level may be beneficial. However, to increase the chance of survival, cancer cells may either stimulate or inhibit autophagy as they progress. Under the context of pro-survival activity, inducing autophagy can greatly improve the proliferation and survival of HCC cell, whereas pro-death autophagy inhibits the progression of tumors. Utilizing nanostructures in the treatment of HCC and autophagy modulation presents a promising avenue, as demonstrated by ongoing research on various NP formulations. Additionally, understanding the regulatory roles of specific miRNAs and advancing genetic tools such as the CRISPR/Cas9 system offer opportunities for more targeted and effective autophagy-based therapies in the future. The current state of HCC treatment is complex because clinicians confront many challenges. Despite the lack of specific and sensitive techniques for early HCC detection, advanced and metastatic HCC diagnosis makes treatment difficult, especially due to medication resistance. Autophagy can anticipate tumor cell response before therapy, allowing for a more successful treatment plan. Since gene therapy is new to cancer treatment, autophagy factors may soon be targeted and manipulated to improve hepatocellular carcinoma.

## Figures and Tables

**Figure 1 cells-13-01186-f001:**
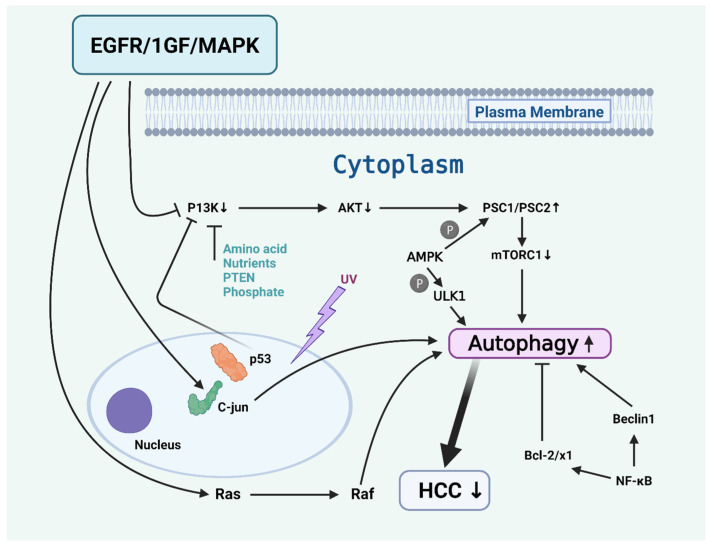
The signaling cascades of autophagy in hepatocellular carcinoma cells (HCCs). The EGFR/IGF/MAPK signaling pathway downregulates the PI3K/mTORC1 pathway and upregulates the C-Jun and Ras/Raf pathways. On the other hand, the AMPK pathway induces autophagy through the phosphorylation of ULK1 and PSC1/PSC2. The NF-κB pathway induces autophagy by activating the Beclin1 and repressing the Bcl-2/X1. The P53 gene also activates autophagy and inhibits cell proliferation of HCC. Figure made with BioRender.com.

**Figure 2 cells-13-01186-f002:**
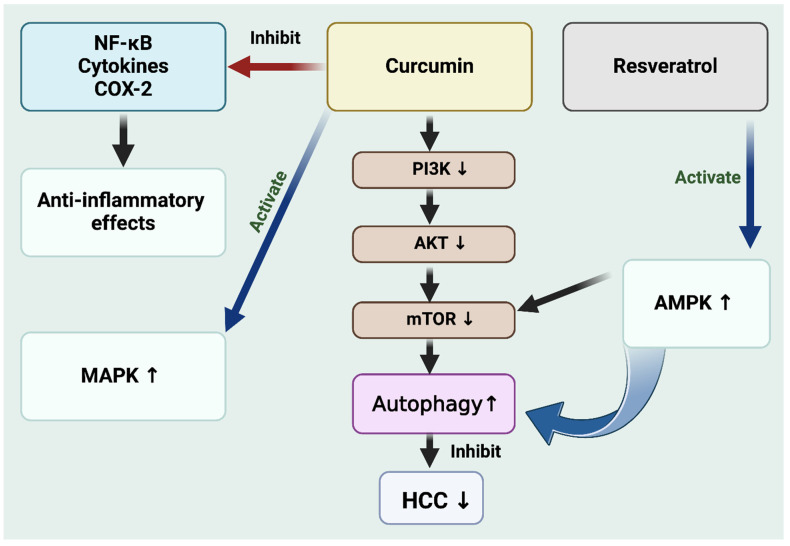
Curcumin and resveratrol inhibit the (PI3K/AKT/mTOR) signaling pathway, increase autophagy, and finally inhibit hepatocellular carcinoma cell proliferation. Similarly, curcumin and resveratrol activate MAPK and AMPK pathways and inhibit HCC. Figure made with BioRender.com.

**Figure 3 cells-13-01186-f003:**
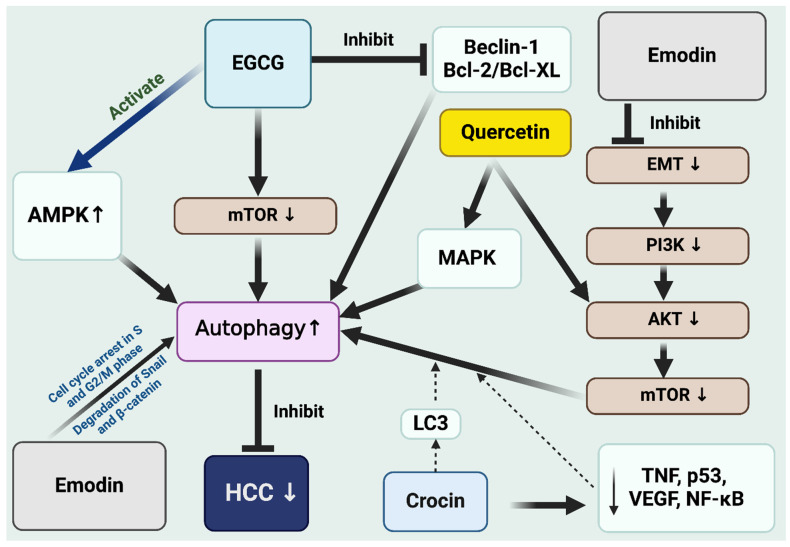
The various natural compounds (EGCG, emodin, quercetin and crocin) inhibit HCC through interconnected signaling pathways. Figure made with BioRender.com.

**Figure 4 cells-13-01186-f004:**
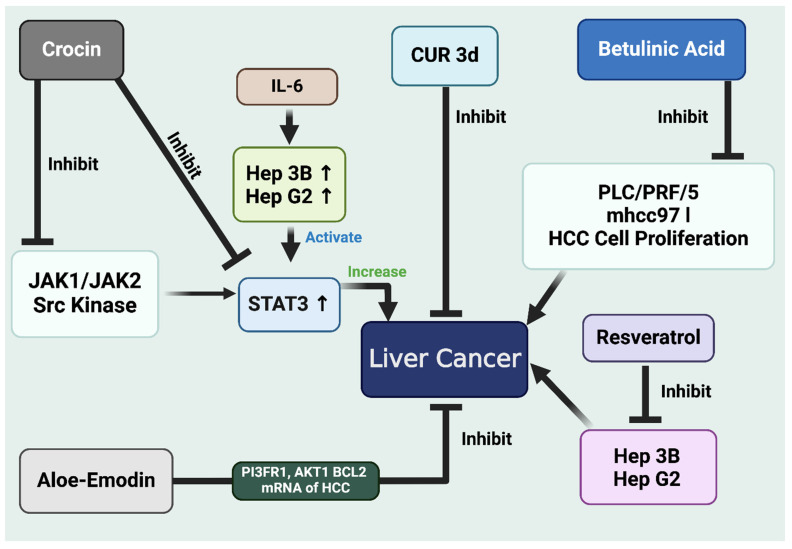
Advanced molecular mechanism of natural compounds related to liver cancer. Crocin, resveratrol, and CUR 3d inhibit liver cancer by exterminating Hep3B and HepG2 cells (cancer cells in the liver). Aloe-emodin inhibits cancer cell formation by downregulating PI3FR1, AKT1, and BCL2 expression, and betulinic acid targets the destruction of cancer cells in the liver by diminishing PLC/PRF/5, MHCC97I, and HCC cell proliferation. Figure made with BioRender.com.

**Table 1 cells-13-01186-t001:** Summary of natural compounds involved in autophagy-modulated liver cancer.

Compounds	Source	Dosage	Administration Method	Mechanism of Action	Potential Side Effects	References
Curcumin	Turmeric	100–200 mg/kg	Oral	Inhibits mTOR signaling	Gastrointestinal discomfort	[93]
Resveratrol	Grapes	10–100 mg/kg	Oral	Activates AMPK pathway	Headache, dizziness	[94]
Berberine	Goldenseal	5–50 mg/kg	Oral	Induces autophagic cell death	Nausea, constipation	[95]
Quercetin	Onions	25–50 mg/kg	Oral	Inhibits PI3K/Akt pathway	Headache, upset stomach	[96]
Epigallocatechin	Green Tea	50–200 mg/kg	Oral	Modulates Beclin-1 and Bcl-2	Liver toxicity at high doses	[97]
Genistein	Soybeans	10–50 mg/kg	Oral	Inhibits Akt/mTOR signaling	Gastrointestinal issues	[98]
Lycopene	Tomatoes	10–50 mg/kg	Oral	Induces autophagy via AMPK	Low toxicity, rare allergic reactions	[99]
Apigenin	Parsley	10–40 mg/kg	Oral	Inhibits PI3K/Akt pathway	Mild gastrointestinal discomfort	[100]
Baicalein	Scutellaria	10–50 mg/kg	Oral	Modulates autophagy and apoptosis	Nausea, vomiting	[101]
Honokiol	Magnolia	5–50 mg/kg	Oral, intravenous	Activates AMPK and inhibits mTOR	Sedation, drowsiness	[102]
Silibinin	Milk Thistle	100–300 mg/kg	Oral	Inhibits mTOR and activates AMPK	Mild gastrointestinal issues	[103]
Withaferin A	Ashwagandha	5–20 mg/kg	Oral	Modulates p62/SQSTM1 pathway	Nausea, skin rash	[104]
Emodin	Rhubarb	10–40 mg/kg	Oral	Inhibits PI3K/Akt/mTOR signaling	Diarrhea, abdominal pain	[105]
Diosgenin	Fenugreek	10–50 mg/kg	Oral	Inhibits Akt/mTOR signaling	Mild gastrointestinal discomfort	[106]
Plumbagin	Black Walnut	2–10 mg/kg	Oral	Induces autophagy via ROS	Hemolysis, nephrotoxicity	[107]
Ursolic Acid	Apple Peel	10–50 mg/kg	Oral	Inhibits mTOR signaling	Mild gastrointestinal discomfort	[108]
Fisetin	Strawberries	10–50 mg/kg	Oral	Inhibits PI3K/Akt/mTOR pathway	Mild gastrointestinal issues	[109]
Luteolin	Celery	10–50 mg/kg	Oral	Inhibits Akt/mTOR signaling	Mild gastrointestinal discomfort	[110]
Ginsenoside Rg3	Ginseng	5–30 mg/kg	Oral, intravenous	Inhibits mTOR and induces autophagy	Mild gastrointestinal discomfort	[111]
Capsaicin	Chili Peppers	2–10 mg/kg	Oral	Activates AMPK pathway	Gastrointestinal irritation

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
