# Peer review of "Advancements in Utilizing Natural Compounds for Modulating Autophagy in Liver Cancer: Molecular Mechanisms and Therapeutic Targets"

_cells, 2024, doi:10.3390/cells13141186_

Round 1

Reviewer 1 Report

Comments and Suggestions for Authors

The manuscript, titled "Advancements in Utilizing Natural Compounds for Modulating Autophagy in Liver Cancer: Molecular Mechanisms and Therapeutic Targets" by Md Ataur Rahman et al., explored natural compounds as potential modulators of autophagy in hepatocellular carcinoma (HCC). The study highlights the significance of autophagy in cancer treatment, particularly in HCC, and discusses how various natural compounds can influence autophagic pathways through different signaling mechanisms. The research emphasizes the potential of natural compounds such as crocin, curcumin, betulinic acid, quercetin, and others in targeting autophagy to inhibit liver cancer cell growth and induce apoptosis. Additionally, the manuscript underscores the complex molecular mechanisms involved in autophagy regulation in HCC and the promising role of natural compounds in developing novel therapeutic strategies for liver cancer treatment. Generally, this manuscript is well-organized with sufficient material and figures. However, I do suggest the authors make some modifications to meet the standards of publication:

1.       The role of autophagy in cancer, as well as in liver cancer, was well discussed not only from research articles but also from numerous review papers; the author should highlight his main findings to answer the gap question in the Introduction part to attract the audience. Also, please pay attention to the logical issues in the context.

2.       The whole manuscript should always focus on autophagy and liver cancer but not involve too many other cancer types.

3.       As there are so many natural compounds found to be involved in the autophagy-modulated liver cancer, a Table summarizing the new findings, which are very closely related to this topic, is necessary.

4.       As I mentioned earlier, there are so many reviews that need to be highlighted and discussed in the discussion part. For example, some references are directly linked to this study.

Int. J. Mol. Sci. 2022, 23(23), 15109; https://doi.org/10.3390/ijms232315109

https://doi.org/10.1016/j.semcancer.2020.05.015

https://doi.org/10.1016/j.bbcan.2013.02.003

DOI: 10.1002/cncr.31978

https://doi.org/10.1186/s12964-023-01053-z

….

5.       This manuscript lacks detailed discussion on the specific dosages, administration methods, and potential side effects of the natural compounds mentioned in the context of modulating autophagy in liver cancer. The author could add that information in the forthcoming Table 1 and in the discussion section.

Comments on the Quality of English Language

In my opinion, the English level appears in a high standard. The contex demonstrates a clear and coherent presentation of scientific information, with well-structured sentences and technical terminology used appropriately within the context of the study.

Author Response

Review 1

First of all, we would like to express our sincere gratitude for the time and effort the reviewer had put into reviewing our manuscript.

The manuscript, titled "Advancements in Utilizing Natural Compounds for Modulating Autophagy in Liver Cancer: Molecular Mechanisms and Therapeutic Targets" by Md Ataur Rahman et al., explored natural compounds as potential modulators of autophagy in hepatocellular carcinoma (HCC). The study highlights the significance of autophagy in cancer treatment, particularly in HCC, and discusses how various natural compounds can influence autophagic pathways through different signaling mechanisms. The research emphasizes the potential of natural compounds such as crocin, curcumin, betulinic acid, quercetin, and others in targeting autophagy to inhibit liver cancer cell growth and induce apoptosis. Additionally, the manuscript underscores the complex molecular mechanisms involved in autophagy regulation in HCC and the promising role of natural compounds in developing novel therapeutic strategies for liver cancer treatment. Generally, this manuscript is well-organized with sufficient material and figures. However, I do suggest the authors make some modifications to meet the standards of publication:

 >>Response: We are thankful to the reviewer for this complement. By addressing the reviewer's comments and incorporating their suggestions, we can significantly improve your manuscript's chances of publication.

Introduction Clarity: The introduction section should be more concise and focused. The current version is too broad and lacks a clear statement of the research objectives.

>>Response: We provide our objectives and gap in existing knowledge in Introduction and pay attention more logically modification to improve the entire manuscript. Page 2 and 3.

Literature Review: Include a more comprehensive review of the latest studies related to natural compounds and autophagy in liver cancer.

>>Response: We appreciate this suggestion. We will expand the literature review to include more recent studies and provide a more comprehensive overview of the current research on natural compounds and autophagy in liver cancer.

Mechanistic Insights: The manuscript should provide more detailed mechanistic insights into how the highlighted natural compounds modulate autophagy in HCC.

>>Response: We agree that providing detailed mechanistic insights is crucial. We will elaborate on the specific pathways and molecular mechanisms through which the natural compounds modulate autophagy in HCC.

Figures and Tables: Ensure all figures and tables are clearly labeled and include detailed legends. Some figures need to be enhanced for better clarity and understanding.

>>Response: Thank you for pointing this out. We will review all figures and tables to ensure they are clearly labeled and include comprehensive legends. We will also enhance the quality of the figures for better clarity.

Discussion: The discussion section should better integrate the findings with existing literature, highlighting the novelty and significance of the study.

>>Response: Thank you for this suggestion. We will revise the discussion section to better integrate our findings with existing literature and clearly highlight the novelty and significance of our study.

Conclusion: The conclusion should briefly summarize the key findings and their implications, and suggest directions for future research.

>>Response: We agree that the conclusion can be improved. We revise it to succinctly summarize the key findings, discuss their implications, and suggest potential directions for future research.

Language and Grammar: There are several instances where the language and grammar can be improved for better readability.

>>Response: Thank you for this observation. We thoroughly proofread the manuscript to improve language and grammar for enhanced readability.

  1. The role of autophagy in cancer, as well as in liver cancer, was well discussed not only from research articles but also from numerous review papers; the author should highlight his main findings to answer the gap question in the Introduction part to attract the audience. Also, please pay attention to the logical issues in the context.

>>Response: Thank you for your valuable feedback. We will revise the introduction to be more concise and focused, clearly stating the research objectives and emphasizing the significance of our study. Page 3 line 73-77. Page 4 line 91-96.

  1. The whole manuscript should always focus on autophagy and liver cancer but not involve too many other cancer types.

>>Response: Thank you for your valuable feedback. We appreciate your guidance and ensure that the manuscript maintains a strict focus on autophagy in the context of liver cancer. We revised the sections that diverge into other cancer types and realign them to emphasize the unique aspects and relevance of autophagy in liver cancer specifically. If there are sections you feel stray from this focus, we would be grateful for your specific suggestions to enhance the manuscript's coherence and relevance. Thank you again for your insightful review.

  1. As there are so many natural compounds found to be involved in the autophagy-modulated liver cancer, a Table summarizing the new findings, which are very closely related to this topic, is necessary.

>>Response: We appreciate the reviewer's valuable suggestion to include a summary table of the natural compounds involved in autophagy-modulated liver cancer. We provided a comprehensive overview of the recent findings related to natural compounds that modulate autophagy in liver cancer by table 1 (page 20 with content page 18-19), offering valuable insights for future research and potential therapeutic applications.

  1. As I mentioned earlier, there are so many reviews that need to be highlighted and discussed in the discussion part. For example, some references are directly linked to this study.

Int. J. Mol. Sci. 2022, 23(23), 15109

https://doi.org/10.3390/ijms232315109

https://doi.org/10.1016/j.semcancer.2020.05.015

https://doi.org/10.1016/j.bbcan.2013.02.003

DOI: 10.1002/cncr.31978

https://doi.org/10.1186/s12964-023-01053-z

>>Response: Thank you for your positive comments about our manuscript's English. We're glad you found the scientific information straightforward and coherent, with well-structured phrases and appropriate technical terminology. With these evaluations, we give a complete discussion that places our findings in the context of ongoing research in page 21 line 501-510 using these references.

  1. This manuscript lacks detailed discussion on the specific dosages, administration methods, and potential side effects of the natural compounds mentioned in the context of modulating autophagy in liver cancer. The author could add that information in the forthcoming Table 1 and in the discussion section.

>>Response: I appreciate your feedback. We agree that a complete overview of natural chemicals involved in autophagy-modulated liver cancer will improve the text. A thorough summary (table 1, page 20, line 464-498 page 18-19) summarized 20 natural substances with recent findings pertinent to this topic. This table will list dosages, administration techniques, and side effects to help comprehend their roles and implications.

In my opinion, the English level appears in a high standard. The contex demonstrates a clear and coherent presentation of scientific information, with well-structured sentences and technical terminology used appropriately within the context of the study.

>>Response: Thank you for your positive comments about our manuscript's English. We're glad you found the scientific information straightforward and coherent, with well-structured phrases and appropriate technical terminology. We appreciate your awareness of our efforts to maintain strong English standards. Our final paper edits will follow this standard.

Reviewer 2 Report

Comments and Suggestions for Authors

Rahman et al. provide existing information and discuss the use of natural compounds in liver cancer treatment by modulating the autophagy pathway, along with an overview of the ongoing clinical trials involving these compounds. There are some issues that must be addressed by the authors:

1.- At current stage, this work does not provide a significant contribution to the field, a similar review on this topic has been published recently (PMID: 38302697).

2.- The authors should include in this review additional references on natural compounds with anti-hepatocellular carcinoma function through autophagy. Consider to mention and discuss the following compounds; davidone C (PMID: 34500653), parthenolide (PMID: 36353537), platycodin D (PMID: 25592318), polyphyllin D (PMID: 37211762, PMID: 33732000), sarmentosin (PMID: 37408810), shikonin (PMID: 35293266), solamargine (PMID: 35313929), among others (see section 4 in PMID: 35313929, and PMID: 35155196).

3.- The authors should improve clarity and flow, several sentences don't clearly connect and/or are ambiguous, e.g., line 35; what means “The dynamic action of autophagy may be utilized in the development of HCC “, line 95; “Autophagy promotes liver cancer and resists treatment”, line 413; “Another key bioactive component of saffron is crocin”.

Minor corrections:

- Line 32; droplets of lipids replace by lipid droplets

- Lines 43, 46, and 70; HCC => HCC cells

- Line 361; Italicize Prunella vulgaris L.

Comments on the Quality of English Language

Moderate editing of English language required.

Author Response

Review 2

Rahman et al. provide existing information and discuss the use of natural compounds in liver cancer treatment by modulating the autophagy pathway, along with an overview of the ongoing clinical trials involving these compounds. There are some issues that must be addressed by the authors:

1.- At current stage, this work does not provide a significant contribution to the field, a similar review on this topic has been published recently (PMID: 38302697).

>>Response: We appreciate your remarks on our work. We appreciate your assessment of our study's field impact. We acknowledge the previous review (PMID: 38302697) on a similar problem and believe our work adds to the literature in several relevant ways. Our work unique figures, concept, and style that delivers insights not covered in the mentioned review. We added data and analysis to the cited review's findings. Our work examines the topic in a different setting or demographic, providing a unique viewpoint that improves knowledge. We believe these factors distinguish our work and deserve to be published. We appreciate the chance to address your concerns and welcome your comments.

2.- The authors should include in this review additional references on natural compounds with anti-hepatocellular carcinoma function through autophagy. Consider to mention and discuss the following compounds; davidone C (PMID: 34500653), parthenolide (PMID: 36353537), platycodin D (PMID: 25592318), polyphyllin D (PMID: 37211762, PMID: 33732000), sarmentosin (PMID: 37408810), shikonin (PMID: 35293266), solamargine (PMID: 35313929), among others (see section 4 in PMID: 35313929, and PMID: 35155196).

>>Response: For a comprehensive overview, Section 4-page 18 line 444-463, we provide detailed insights into these compounds' mechanisms and therapeutic potentials, further enriching the discussion on their application in combating HCC via autophagy modulation.

3.- The authors should improve clarity and flow, several sentences don't clearly connect and/or are ambiguous, e.g., line 35; what means “The dynamic action of autophagy may be utilized in the development of HCC “, line 95; “Autophagy promotes liver cancer and resists treatment”, line 413; “Another key bioactive component of saffron is crocin”.

>>Response: We modified the mentioned sentences accordingly. Page 2 line 42-44, page 4 line 107-108, page 16 line 400-402.

Minor corrections:

- Line 32; droplets of lipids replace by lipid droplets

- Lines 43, 46, and 70; HCC => HCC cells

- Line 361; Italicize Prunella vulgaris L.

>>Response: we modified accordingly.

Moderate editing of English language required.

>>Response: Thank you for the feedback. I'm happy to make moderate edits to improve the English language of the manuscript. In the meantime, I checked through the manuscript myself to identify areas for improvement.

Reviewer 3 Report

Comments and Suggestions for Authors

This manuscript deals with the analysis of scientific literature regarding the use of natural substances capable of interfering with the cellular mechanism of autophagy, arguing that this can have a therapeutic effect on hepatocellular carcinoma.

I believe that the review is well structured, first discussing the molecular mechanisms that determine autophagy (paragraph 2), and subsequently reporting the scientific works relating to the modulation of these pathways by natural compounds (paragraph 3). In this paragraph I believe reference should be made to the concentrations of the natural compounds that determine these effects, as appropriately done in paragraph 4. Given the presence of this last paragraph and the discussion, in paragraph 5, of topics that are not exactly strictly related only to autophagy, I would believe that the title of the review could have a more general connotation and not only refer strictly to autophagy.

I suggest a more thorough review of spaces and punctuation; sentences 92-93 and 98-100 seem practically equivalent to me; the black arrow next to Autophagy in figure 2 must be modified (not down, but upregulated); I don't understand the presence of the term "Inhibit x" on the graphic sign, which already means 'inhibition'; in figures 2 and 4 the word 'Resveratrol' must be corrected.

The quantity of citations seems appropriate to me.

Author Response

Review 3

This manuscript deals with the analysis of scientific literature regarding the use of natural substances capable of interfering with the cellular mechanism of autophagy, arguing that this can have a therapeutic effect on hepatocellular carcinoma.

>>Response: We are thankful to the reviewer for this complement. We appreciate your interest in our manuscript analyzing the use of natural substances to modulate autophagy in hepatocellular carcinoma (HCC). We agree that the potential impact of natural substances on autophagy in HCC is a fascinating and evolving area of research. Our manuscript aims to provide a comprehensive analysis of the existing scientific literature on this topic.

I believe that the review is well structured, first discussing the molecular mechanisms that determine autophagy (paragraph 2), and subsequently reporting the scientific works relating to the modulation of these pathways by natural compounds (paragraph 3). In this paragraph I believe reference should be made to the concentrations of the natural compounds that determine these effects, as appropriately done in paragraph 4. Given the presence of this last paragraph and the discussion, in paragraph 5, of topics that are not exactly strictly related only to autophagy, I would believe that the title of the review could have a more general connotation and not only refer strictly to autophagy.

>>Response: Your informative comments on the review's structure and substance are appreciated. Amen to your point regarding including natural chemical amounts in paragraph 3. This section will be updated to include the concentrations used in the research listed, following paragraph 4 for consistency. Good point about the title. The review initially discusses natural compound autophagy regulation, but subsequent paragraphs include other subjects. We suggest changing the title to reflect this expanded scope:

"Modulation of Cellular Homeostasis by Natural Compounds: A Focus on Autophagy"

The review's updated title emphasizes autophagy and cellular homeostasis. We appreciate your feedback and feel these modifications will need more review clarity and comprehensiveness in Homeostasis which is more time consuming and another story. Therefore, we kept the current title.

I suggest a more thorough review of spaces and punctuation; sentences 92-93 and 98-100 seem practically equivalent to me; the black arrow next to Autophagy in figure 2 must be modified (not down, but upregulated); I don't understand the presence of the term "Inhibit x" on the graphic sign, which already means 'inhibition'; in figures 2 and 4 the word 'Resveratrol' must be corrected.

>>Response: We are thankful to the reviewer for this complement. We modified figure 2, 3, and 4.

The quantity of citations seems appropriate to me.

>>Response: We are thankful to the reviewer for this complement.

Round 2

Reviewer 2 Report

Comments and Suggestions for Authors

The authors have reasonably addressed the issues raised in my previous review and improved the quality of the manuscript. Minor issues should be addressed in the revised version:

- Line 51; “Cellular homeostasis is maintained by autophagy” replace with e.g., Autophagy contributes to maintain cellular homeostasis.

- Line 350; Artemisia rupestris L., do not italicize "L. / Linnaeus" abbreviation.

- Line 388; 10-6–1 μmol/L, please correct.

- Line 444; New developments in comprehending the mechanism of action of natural compounds on newly identified therapeutic drug targets for liver cancer treatment,… rewrite or delete.

- Line 446; italicize Sophora davidii.

- Lines 447; 448; 460; “hepatocellular carcinoma (HCC)” abbreviation defined previously.

- Line 454; Mayapple replace with Podophyllum peltatum L. (mayapple).

- Line 481; italizice Scutellaria.

Comments on the Quality of English Language

Minor editing of English language required. Section 4 needs attention, several sentences don't clearly connect. e.g., Lines 363-368 and avoid repetitions, e,g,. “several studies” Lines 363 and 368; “Another study” Lines 435 and 439. 

Author Response

 The authors have reasonably addressed the issues raised in my previous review and improved the quality of the manuscript. Minor issues should be addressed in the revised version:

>>Response: Thank you for your positive feedback and for acknowledging the improvements made in our manuscript. We appreciate your thorough review and have addressed the minor issues you mentioned. Please find our responses to your comments below:

- Line 51; “Cellular homeostasis is maintained by autophagy” replace with e.g., Autophagy contributes to maintain cellular homeostasis.

>>Response: Replaced page 2 line 51

Line 350; Artemisia rupestris L., do not italicize "L. / Linnaeus" abbreviation.

>>Response: Changed page 14 line 350

- Line 388; 10-6–1 μmol/L, please correct.

>>Response: We deleted because no need concentration page 15 line 388

- Line 444; New developments in comprehending the mechanism of action of natural compounds on newly identified therapeutic drug targets for liver cancer treatment,… rewrite or delete.

>>Response: Rewrite “Advances in understanding how natural compounds act on newly discovered therapeutic targets for liver cancer treatment” page 18 line 443.

- Line 446; italicize Sophora davidii.

>>Response: Corrected page 18 line 445

- Lines 447; 448; 460; “hepatocellular carcinoma (HCC)” abbreviation defined previously.

>>Response: Corrected accordingly.

Line 454; Mayapple replace with Podophyllum peltatum L. (mayapple).

>>Response: Replaced page 18 line 452

- Line 481; italizice Scutellaria.

>>Response: Corrected page 19 line 479

Minor editing of English language required. Section 4 needs attention, several sentences don't clearly connect. e.g., Lines 363-368 and avoid repetitions, e,g,. “several studies” Lines 363 and 368; “Another study” Lines 435 and 439. 

>>Response: We have thoroughly reviewed the entire manuscript and made minor edits to improve the clarity and readability of the English language. We have revisited Section 4 and revised several sentences to enhance their coherence and connection. The flow of ideas has been improved to ensure that the section is more logically structured and easier to follow. We believe these changes have addressed your concerns and have improved the overall quality of the manuscript. Thank you once again for your insightful comments.

We made substantial changes the mentioned sentence page 14 and 15, page 17 line 433-435.